# Graft–Host Interaction and Its Effect on Wound Repair Using Mouse Models

**DOI:** 10.3390/ijms242216277

**Published:** 2023-11-13

**Authors:** Nicole Garcia, Md Mostafizur Rahman, Carlos Luis Arellano, Ilia Banakh, Chen Yung-Chih, Karlheinz Peter, Heather Cleland, Cheng Hean Lo, Shiva Akbarzadeh

**Affiliations:** 1Skin Bioengineering Laboratory, Victorian Adult Burns Service, Alfred Health, 89 Commercial Road, Melbourne, VIC 3004, Australia; nicole_garcia@me.com (N.G.); mostafiz.rahman@monash.edu (M.M.R.); carlos.arellano1@monash.edu (C.L.A.); ilia.banakh@monash.edu (I.B.); h.cleland@alfred.org.au (H.C.); cheng.lo@monash.edu (C.H.L.); 2Department of Surgery, Monash University, 99 Commercial Road, Melbourne, VIC 3004, Australia; 3Atherothrombosis and Vascular, Baker Heart and Diabetes Institute, Melbourne, VIC 3004, Australia; yungchih.chen@baker.edu.au (C.Y.-C.); karlheinz.peter@baker.edu.au (K.P.)

**Keywords:** IL-6, TGF-β1, wound repair, skin grafting, myofibroblast

## Abstract

Autologous skin grafting has been commonly used in clinics for decades to close large wounds, yet the cellular and molecular interactions between the wound bed and the graft that mediates the wound repair are not fully understood. The aim of this study was to better understand the molecular changes in the wound triggered by autologous and synthetic grafting. Defining the wound changes at the molecular level during grafting sets the basis to test other engineered skin grafts by design. In this study, a full-thickness skin graft (SKH-1 hairless) mouse model was established. An autologous full-thickness skin graft (FTSG) or an acellular fully synthetic Biodegradable Temporising Matrix (BTM) was grafted. The wound bed/grafts were analysed at histological, RNA, and protein levels during the inflammation (day 1), proliferation (day 5), and remodelling (day 21) phases of wound repair. The results showed that in this mouse model, similar to others, inflammatory marker levels, including *Il-6*, *Cxcl-1*, and *Cxcl-5*/*6*, were raised within a day post-wounding. Autologous grafting reduced the expression of these inflammatory markers. This was different from the wounds grafted with synthetic dermal grafts, in which *Cxcl-1* and *Cxcl-5/6* remained significantly high up to 21 days post-grafting. Autologous skin grafting reduced wound contraction compared to wounds that were left to spontaneously repair. Synthetic grafts contracted significantly more than FTSG by day 21. The observed wound contraction in synthetic grafts was most likely mediated at least partly by myofibroblasts. It is possible that high TGF-β1 levels in days 1–21 were the driving force behind myofibroblast abundance in synthetic grafts, although no evidence of TGF-β1-mediated Connective Tissue Growth Factor (CTGF) upregulation was observed.

## 1. Introduction

In humans, spontaneous wound repair proceeds in three overlapping phases of inflammation (0–10 days), proliferation (2–30 days), and remodelling (10–300 days). These phases are largely defined at the cellular and molecular levels. This process is controlled by a balance of growth factors expressed in a spatial and temporal fashion [1,2,3]. In the first stage of acute wound repair, vasoconstriction and clotting cascade are initiated by the platelets. They form a provisional matrix acting as a scaffold for the leukocyte migration [4,5]. Both platelets and leukocytes release cytokines and growth factors to activate the inflammatory process (interleukins (IL): IL-1α, IL-1β, and IL-6 and tumour necrosis factor alpha (TNF-α)); stimulate collagen synthesis (fibroblast growth factor 2 (FGF-2), insulin-like growth factor 1 (IGF-1), transforming growth factor beta 1 (TGF-β1)); activate the transformation of fibroblasts to myofibroblasts (TGF-β1); commence angiogenesis (FGF-2, VEGF-A, HIF-1α, TGF-β1); and support the re-epithelialisation process (EGF, FGF-2, IGF-1, TGF-α) [6]. IL-8 is also released during the inflammation phase. Rodents lack a direct homologue of IL-8; however, KC (keratinocyte-derived chemokine, CXCL-1) and LIX (lipopolysaccharide-induced CXC chemokine, CXCL-5/6) are considered functional homologues due to their predominant involvement in neutrophil recruitment [7].

Fibroblast proliferation, epithelialisation, and angiogenesis begin shortly after wounding. Platelets and macrophages from the haemostatic clot continue to stimulate fibroblast proliferation and migration into the wound bed by producing platelet-derived growth factor (PDGF) and TGF-β1. By the third or fourth day, the wound becomes rich in fibroblasts, which lay down the extracellular matrix (ECM)—mainly collagen III (Col III) and other ECM proteins, such as Col I, hyaluronan, fibronectins, and proteoglycans. Nearby damaged endothelial cells secrete basic fibroblast growth factor (or FGF-2) to signal endothelial cell migration [8]. Fibroblasts then synthesise and secrete keratinocyte growth factor 1 (KGF-1), keratinocyte growth factor 2 (KGF-2), and IL-6 to stimulate adjacent keratinocytes’ migration to the wound area, proliferation, and differentiation in the neo-epidermis [2]. The concurrent formation of new blood vessels (angiogenesis) is necessary to sustain the newly formed granulation tissue. This relies on ECM deposition in the wound bed, as well as the stimulation of endothelial cells by vascular endothelial growth factor (VEGF) to form neo-dermis [5]. VEGF is secreted predominantly by keratinocytes on the wound edge, but also by macrophages, platelets, fibroblasts, and endothelial cells in response to injury and hypoxia [2]. Fibroblasts from the wound edge and the bone marrow may begin transforming into myofibroblasts under the influence of macrophage-secreted TGF-β1, whilst synthesising collagen. The appearance of contractile myofibroblasts corresponds to the commencement of wound contraction [9].

Types I and III collagens make up about 95% of the scar collagen. In uninjured skin, collagen consists of 80–90% Col I and 10–20% Col III; granulation tissue has 30% Col III; whereas a mature scar only has 10% Col III [10]. This continual decrease in Type III collagen within maturing scar is mediated by the matrix remodelling proteinases (MMPs), secreted by fibroblasts in the neo-dermis under the influence of TGF-β1, PDGF, IL-1, and EGF. The MMPs, in turn, are controlled by tissue inhibitors of metalloproteinase (TIMPs), which are also upregulated in fibroblasts by TGF-β1 and IL-6. The balance between MMPs and TIMPs activity in the neo-dermis determines the remodelling outcome.

In wounds with extensive skin loss, spontaneous repair is not achievable in a timely manner and skin grafting is required to avoid excessive granulation and subsequent scarring [11]. In skin graft animal models, when a wound is grafted, adherence and plasmatic imbibition phases commence immediately on the first day. The graft provides ECM, which is mostly collagen and which re-vascularises in 2–5 days. The graft remodelling phase starts within 7 days post-grafting and can last for weeks [12,13,14]. More recent studies have suggested that new skin under a split-thickness skin graft in pigs regenerates in a “bottom-up” process by the appearance of α-SMA-positive expressing cells in deeper layers [15]. In general, animal studies have been largely interested in the vascularisation of grafts with little attention paid to the molecular changes during this process, which drives the graft take. However, MMPs, particularly MMP-2, MMP-9, and MT1-MMP (that control ECM remodelling during spontaneous wound repair), have been identified as the main proteases driving vasculature and ECM remodelling in grafts [16,17].

The limited availability of donor skin has been the driving force in developing biological or synthetic dermal substitutes to reconstitute the cutaneous dermal layer [18]. Dermal substitutes act as matrices or scaffolds, promoting new tissue growth and enhancing wound repair [19,20], possibly improving the resultant scar quality [21]. The emergence of engineered dermal substitutes in clinical use as an adjunct to autologous skin grafts has reinvigorated the need to understand molecular changes in the neo-dermis post-grafting. This knowledge will contribute to establishing requirements when designing novel dermal substitutes that vascularise readily and provide long-term robust healing.

Biodegradable Temporising Matrix (BTM) is a fully synthetic dermal substitute in the form of biodegradable polyurethane foam with a temporary non-biodegradable polyurethane seal. BTM and autologous FTSG were grafted on splinted full-thickness wounds in SKH-1 mice. The aims of this study were (i) to analyse molecular changes in the wound microenvironment in autologous skin grafts over time and (ii) to compare the molecular changes in autologous skin grafts to BTM grafts and examine their consequences for wound repair. To our knowledge, the study reported herein is the first to establish the growth factor trajectory of wound repair through both autologous and synthetic dermal grafts.

## 2. Results

### 2.1. The Splinted Full-Thickness Wound Repair in SKH-1 Mice Mimics the Stages of Spontaneous Wound Repair in Humans

In order to study the effect of grafting on the trajectory of spontaneous wound repair, first, the splinted full-thickness wounds were allowed to spontaneously heal over 21 days in a mouse model via epithelisation (Appendix A). Granulation tissue was laid down in splinted wounds between days 1 and 8 and the re-epidermisation was evident from day 8 onwards. By day 14, a hyperproliferative neo-epidermis had fully covered the wounds, and by day 21, evidence of wound remodelling was apparent. The abundance of inflammatory cytokines/growth factors (TNF-α, IL-1β, IL-6, and the IL-8 homologues CXCL-1 and CXCL-5/6) in the healing wound were analysed on days 1, 2, 5, 8, 11, 14, and 21, compared to their systemic levels in skin using RT-qPCR (Appendix A). Most inflammatory markers peaked by day 5 in the wounds and showed a second peak by day 21. The proliferative markers VEGF-A, Collagen 3 (Col3), and Collagen 1 (Col1) had increased by day 5. Although wound closure was achieved by day 14, collagen overexpression continued up until 21 days post-wounding. The anti-inflammatory marker IL-10 expression level was generally low throughout the spontaneous wound repair, except a peak observed on day 2, in this model. The wound repair markers’ profile in the splinted SKH-1 mouse model presented here largely aligns with stages of spontaneous wound repair in humans [1,2].

### 2.2. Grafting Reduces Wound Contraction

Two splinted full-thickness wounds were created on the back of SKH-1 mice (Figure 1). The mice were grafted with either autologous FTSG or synthetic dermal on one side and left ungrafted to spontaneously repair on the other side. The mice were monitored for 21 days. The ungrafted wounds were significantly contracted compared to the day 0 wounds from day 10 onwards (*p* < 0.0001). The skin contraction continued till the D21 endpoint (*p* < 0.0001). Autologous skin grafting inhibited wound contraction compared to when the wounds were left open to repair spontaneously. Although grafting with the synthetic dermal substitute, BTM, reduced wound contraction, compared to ungrafted wounds, it was not as effective as FTSG. FTSGs remained similar in size over 21 days post-grafting, whereas the BTM grafts contracted significantly over this period (day 5 *p* = 0.0009, day 10 *p* < 0.0001, day 15 *p* < 0.0001) (Figure 1).

### 2.3. Autologous Skin Grafting Dampens Overexpression of Inflammatory Il-6, Cxcl-1, and Cxcl-5/6 in the Wound

In order to study the effect of autologous skin grafting (FTSG) on the wound inflammation, *Il-6*, *Cxcl-1*, and *Cxcl-5*/*6* expressions were measured in grafted vs. ungrafted neo-dermis on day 1, day 5, and day 21 post-grafting using RT-qPCR (Figure 2). The contribution of epidermis to the levels of inflammatory markers was eliminated by removing the grafted epidermis prior to RNA purification. The growth factor expression levels in the grafted dermis (or ungrafted wound bed) were normalised against their expression in uninjured dermis near the “tail” on the dorsal side harvested on the same day as the wound/graft. The data show that the transient accumulation of *Il-6* in autologous FTSG is significantly reduced compared to the *Il-6* levels in ungrafted wounds during the inflammation phase on day 1 (*p* = 0.0127). *Il-6* overexpression ceased at both mRNA and protein levels in autologous grafts by day 21. Moreover, the *Cxcl-1* level was significantly lower at the skin-grafted wound on day 1 compared to ungrafted wounds (*p* = 0.002). Similarly, *Cxcl-5*/*6*, possibly secreted by macrophages attracted to the wound site, was significantly lower at the skin-grafted wounds on day 1 (*p* = 0.0028). Consistent with a shorter inflammatory phase in autologous skin grafts, haematoxylin and eosin staining demonstrated fewer inflammatory cells in autologous skin-grafted wounds compared to ungrafted wounds (Figure 3A).

### 2.4. Resolved Inflammation in Autologous Skin Grafts Dampens Col1 and Col3 Expression

To analyse the effect of autologous grafting on the expression of wound repair proliferation markers, the VEGF, *Col1*, and *Col3* levels in the wounds were measured using RT-qPCR (Figure 2). The analysis revealed that Col1 expression had significantly dropped at the grafted side on day 5 (*p* < 0.0001). Moreover, *Col3* expression was reduced at the grafted side by day 5, which reached significance by day 21 (*p* = 0.012). The proliferation marker, *VEGF-A* expression level, remained the same in the grafted vs. ungrafted wound bed.

### 2.5. Synthetic Dermal Grafting Showed a Trend of Extended Inflammation and Proliferation Phases of Wound Repair

To compare the effect of a fully synthetic grafting to FTSG on the neo-dermis, the wounds were grafted with BTM and analysed in a similar fashion to FTSG. In contrast to autologous skin grafting, synthetic grafting seemed to maintain a high local inflammatory state in the grafts for longer, although higher local inflammation did not reach significance (Figure 2). There was a trend in higher levels of *Cxcl-1* and *Cxcl-5/6* in BTM grafts compared to FTSG and ungrafted wounds up to 21 days. *Il-6*, however, was resolved in the BTM grafted wounds, similar to FTSG and ungrafted wounds by day 21 post-grafting. Interestingly, there was an upward trend of anti-inflammatory IL-10 on day 21 in BTM grafts compared to ungrafted wounds. Similarly, there was a trend of *Col1* and *Col3* overexpression in BTM grafts on day 5 and day 21 compared to FTSG. Although the collagen overexpression continued up to day 21, the total amount of dermal collagen in BTM grafts still lagged behind the FTSG over the whole 21 days at protein level (Figure 3). “Older collagen” in the FTSG is denser and it is therefore stained darker blue in Masson trichome staining. In BTM grafts, the darker blue is present at the wound edge and the deeper part of the neo-dermis on days 5 and 21. The granulation tissue stained a lighter blue (freshly deposited collagen) in the middle of the wound and closer to the surface, suggesting that collagen is deposited within the scaffold not only upwards from the wound bed, but also from the wound edges in this mouse model to form neo-dermis.

### 2.6. α-SMA^+^/VIM^+^ Myofibroblasts Contribute to Graft Contraction in Synthetic Dermal Grafts

Double staining of grafts for α-smooth muscle actin (α-SMA) and vimentin (VIM) was employed to detect contractile myofibroblasts in the neo-dermis on days 5 and 21 post-grafting (Figure 4). A low number of VIM^+^ fibroblasts and α-SMA^+^/VIM^+^ myofibroblasts were observed in the FTSG grafts on day 5 and day 21. The number of Vim^+^ fibroblasts was increased in the BTM grafts compared to FTSG on day 21 post-grafting (*p* = 0.0214) and more importantly, the ratio of α-SMA^+^/VIM^+^ myofibroblast to VIM^+^ fibroblast was higher in BTM grafts compared to FTSG. Almost 45% of VIM^+^ fibroblasts had converted to α-SMA^+^/VIM^+^ myofibroblasts in BTM grafts, whereas only 9% of VIM^+^ fibroblasts were converted to α-SMA^+^/VIM^+^ myofibroblasts in FTSG by day 21 (*p* = 0.0274).

### 2.7. TGF-β1 Remains High in Synthetic Grafts

In order to understand the mechanism by which fibroblast to myofibroblast transformation was stimulated in BTM grafts, TGF-β1 levels were measured using RT-qPCR and ELISA in ungrafted wounds, FTSG, and BTM grafts (Figure 5). Autologous grafting reduced TGF-β1 in the neo-dermis at both mRNA and protein levels over time (*p* = 0.0317). Unlike autologous grafting, BTM grafting was not associated with decreased TGF-β1 at either mRNA or protein levels over 21 days. To analyse TGF-β1 downstream signalling, the CTGF levels in ungrafted wounds, FTSG, and BTM grafts were analysed using ELISA. CTGF is a typical TGF-β/Smad response gene [22]. CTGF levels remained similar in all three groups. No evidence of CTGF involvement in wound repair was observed in these grafts.

## 3. Discussion

This study found that the local production of *Il-6*, *Cxcl-1*, *Cxcl-5*/*6*, *VEGF*, and *TGF-β1* growth factors at RNA level in neo-dermis throughout spontaneous wound repair exceeded their levels in native skin. This was achieved by analysing the growth factor levels in the neo-dermis (the wound bed) during the wound repair, separate from the neo-epidermis, and comparing their levels to those in the posterior dorsal skin dermis, away from the wound near the “tail”, retrieved on the same day. The local production of the listed growth factors in autologous and synthetic grafted wounds was analysed to indicate the effect of grafting on the trajectory of the wound repair. When wounds were grafted with autologous skin, the inflammatory *Il-6*, *Cxcl-1*, and *Cxcl-5*/*6* expression in the neo-dermis was significantly reduced at RNA level on day 1, day 5, and day 21 post-grafting compared to ungrafted wounds. At the protein level, inflammatory IL-6 was detected in (the dermis of) the autologous graft on day 1 and day 5, but its level dropped to almost nil by day 21. These observations support the hypothesis that autologous skin grafting dampens the local inflammatory response. The resolution of local inflammation allows the wound to move to the proliferation and remodelling phases and therefore facilitates wound repair. It is not clear what cell type mediates the resolving of local inflammation, although native fibroblasts and endothelial cells in the grafted dermis may play a significant role in this process. Downward signals from grafted keratinocytes can also contribute to inflammation resolution [23,24].

In contrast to spontaneous wound repair (i.e., ungrafted splinted full-thickness wounds), where both *Col1* and *Col3* levels rose during the proliferation phase (day 5), and in autologous grafts, *Col1* and *Col3* expression was dampened by day 5. The likely mechanism for this was that FTSG provides ECM collagen proteins (in the form of the neo-dermis) to the wound, thus shortening fibroblast activation and ECM protein upregulation. Furthermore, in FTSG, the total ECM collagen declined between D5 and D21, implying that some graft remodelling had commenced by day 21.

In comparison to autologous grafting, fully synthetic dermal grafting was associated with longer inflammation and proliferation phases in the wound. For example, *Col1* and *Col3* levels remained high 21 days post-grafting in the BTM grafts, although their total collagen still did not reach the levels of total collagen in FTSG. In addition, greater wound contraction in BTM grafts was observed compared to FTSG over 21 days post-grafting. These observations are consistent with the current clinical acceptance that the ability of a skin graft to inhibit wound contraction is directly proportional to the amount of collagen within the dermis [25]. The structure and appearance of collagen within the granulation tissue changes with time, allowing for the recognition of older versus newer collagen as well as the transition of “parallel-like” immature collagen to the “basket-weave” pattern of collagen in mature normal skin [26]. New collagen deposited in the ungrafted wounds and BTM grafts showed a parallel appearance compared to basket-weave collagen in FTSG.

Myofibroblasts are activated fibroblasts that orchestrate wound repair by depositing high amounts of collagen and by triggering wound contraction. Myofibroblasts gain contractile microfilament apparatuses (normally absent in fibroblasts) that are organised in bundles. Microfilament bundles show an expression of α-SMA [27]. In this study, both SMA and VIM expression in fibroblasts were compared in FTSG and BTM grafts. A significantly greater proportion of VIM^+^ fibroblasts were differentiated to VIM^+^/α-SMA^+^ myofibroblasts in BTM grafts compared to FTSG. It is not clear what factor(s) mediates the increased VIM^+^/α-SMA^+^ myofibroblast differentiation in BTM grafts. The two possibilities are that (1) the myofibroblast differentiation is mediated by the overactivation of an existing pathway in native skin grafting, or (2) the myofibroblast differentiation in BTM grafts occurs via an independent pathway that is novel. More analysis is needed to fully understand the myofibroblast differentiation mechanism.

It is interesting to note that the accumulation of VIM^+^/α-SMA^+^ myofibroblasts in BTM grafts coincided with the prolonged overexpression of TGF-β1. The role of TGF-β1 in wound repair is rather complex. TGF-β1 stimulates granulation tissue formation, including vessel formation and collagen deposition [28,29]. TGF-β1 accelerates wound repair in partial-thickness murine wounds through its stimulatory effect on fibroblasts and ECM deposition. In full-thickness wounds, however, it has been demonstrated that TGF-β1 delays wound closure through its inhibitory effect on keratinocyte migration [30], and prolongs the inflammatory phase through its intrinsic pro-inflammatory effect [31]. Moreover, there is evidence that the prolonged expression of TGF-β1 may lead to aberrant scar formation, as it is a potent stimulator of collagen deposition [32] as well as suppressing the expression of MMPs and therefore reducing ECM degradation by fibroblasts [33]. As a result, TGF-β1 has been implicated in aberrant scar pathologies such as keloids [34]. Given the high level of TGF-β1 in BTM grafts in this study, it would be interesting to measure ECM remodelling enzymes such as MMPs and TIMPs in this model to show a possible correlation.

Autologous grafting is difficult to achieve in the case of large wounds and there is a clinical need to develop synthetic grafts as an adjunct treatment. The idea behind synthetic grafting is that in large surface area cutaneous injuries of differing depths, such as burns, the period in which the sealed matrix integrates into the wound bed and clinically appears to physiologically “close” the wound allows for the repair of other, more superficial wounds, including split skin graft donor sites. This decreases the systemic inflammatory response that characterises massive burns and allows for the stabilisation of the patient prior to definitive skin grafting. It also provides a scaffold to guide the formation of a neo-dermis in a controlled fashion and improves the vascularisation of the wound bed. This study has provided a detailed analysis of changes in molecular mediators in the wound/neo-dermis or wound/FTSG dermis interface during engraftment. Furthermore, it has highlighted the differences in molecular changes associated with synthetic grafting compared with autologous skin grafting on days 1–21. A limitation of this study in comparing a dermal substitute with a FTSG, however, is that unlike FTSG that replaces both dermis and epidermis, the temporary coverage of BTM is acellular and does not contain a functional epidermis. Therefore, this comparison does not account for the effect of the mesenchymal–epidermal interaction that exists in autologous skin grafts. For example, a secreted form of Stratifin from grafted keratinocytes has been shown to stimulate the expression of MMP-1 collagenase in dermal fibroblasts, facilitating ECM remodelling in an FTSG with an effect on reducing long-term scarring [23,24]. Another limitation of the study is that the dermal cell populations in FTSG and BTM grafts were not analysed. FTSGs contribute to neo-dermis cell populations, including endothelial and immune cells, instantly post-grafting, whereas the BTM graft cellularity lags behind FTSG as it totally relies on host cell migration. This should be taken into consideration when interpreting these results. Nevertheless, this study provides a benchmark analysis of a wound environment grafted with autologous skin grafting and a fully synthetic dermal substitute. Further investigation is needed to identify the molecular differences between the autologous skin grafting and dermal substitutes during graft remodelling beyond 21 days.

In conclusion, we have shown how stages of spontaneous wound repair are influenced by autologous skin grafting. Moreover, we have identified significant differences in wound repair pathway activation when wounds are grafted with a synthetic graft compared with autologous FTSG. The long-term goal of this study is to identify biomarkers that can predict healing and scarring outcomes post-grafting. Graft take and scarring biomarkers are useful tools to assist in novel and efficacious engineered graft designs and/or graft outcomes.

## 4. Materials and Methods

### 4.1. Ethics

All experiments were performed on animals in accordance with the guidelines set by the Animal Ethics Committee (AEC) of Monash University and Baker Institute, Melbourne (AEC E/1920/2019/A) and in accordance with National Institutes of Health guide for the care and use of laboratory animals (NIH Publications No. 8023, revised 1978).

### 4.2. Surgical Procedures

Male SKH-1 hairless mice aged 8–10 weeks were anaesthetised as previously described with some modifications [35]. Briefly, two full-thickness wounds, 1.5 cm in diameter, were created in the dorsal skin. Two custom-made silicon rings (Thermo Fisher Scientific, Waltham, MA, USA) were adhered to the outer edges of each wound using skin Histoacryl^®^ glue (B. Braun, Melsungen, Germany) and sutured in place along the outer circumference of the ring. An autologous full-thickness skin graft (FTSG) or the synthetic Biodegradable Temporising Matrix (BTM) was sutured onto one of the two wounds with monofilament nylon sutures (Warner & Webster, North Sydney, Australia) with 4 to 6 simple interrupted sutures. The adjacent wound was left as an ungrafted full-thickness wound. The wounds were dressed with one layer of a paraffin-impregnated gauze (Jelonet, Smith & Nephew, Watford, UK), a non-absorbent wound dressing (Surfasoft^®^, Otago Healthcare, Northampton, UK), and dry gauze. These were fixed in place with an adhesive waterproof dressing (Tegaderm™, 3M Company, St. Paul, MN, USA), and a self-adherent bandage (Coban™, 3M Company). The mice were given intraperitoneal injection of non-opioid analgesia (paracetamol, 3 mg/mL in saline) and opioid analgesia (buprenorphine, 0.01 mg/kg). The mice were euthanised at their assigned end-points using a CO_2_/CO_2_ gas mix.

### 4.3. ELISA

Protein was purified from snap-frozen tissue by homogenising in 3 × 10 s pulses in RIPA buffer. The protein concentrations were determined using a Pierce™ BCA assay (Thermo Fisher Scientific) according to the manufacturer’s instructions. Then, 45 µg protein/well was assessed for mouse IL-6 (BD Bioscience, San Diego, CA, USA), IL-10 (BD Bioscience), TNF-α (Thermo Fisher Scientific), TGF-β (Thermo Fisher Scientific) and CTGF (Elabscience, St. Paul, TX, USA) content using ELISA according to the manufacturer’s instructions.

### 4.4. RT-qPCR

RNA (grafted wound, ungrafted wound, and uninjured skin) was extracted using a RNeasy mini kit (Qiagen, Hilden, Germany) according to the manufacturer’s instructions. cDNA was synthesised using the GoScriptTM Reverse Transcriptase Master Mix and Oligo d(T) kit (Promega, Madison, WI, USA) from 400 ng RNA template according to the following program: 25 °C for 10 min, 42 °C for 60 min, 99 °C for 5 min, 4 °C for 5 min. RT-qPCR was performed as previously described [36]. Briefly, amplification was performed using the GoTaq^®^ RT-qPCR MasterMix (Promega) in a 384-well plate and specific primers for the genes of interest and three reference housekeeping genes. All specific primers spanned over an exon–intron junction (Table 1). mRNA expression levels were calculated using the Comparative Method [37], which uses the C_T_ values to evaluate the fold expression of each target gene. The fold expression is defined as 2^−ΔΔCT^, where –ΔΔC_T_ = ((C_T_ Gene of interest in graft/wound − C_T_ Average of 3 endogenous housekeeping genes in graft/wound) − (C_T_ Gene of interest in tail − C_T_ Average of 3 endogenous housekeeping genes in tail)).

### 4.5. Histochemistry

Skin tissues were fixed in 4% paraformaldehyde prior to cryopreservation. The specimens were stained with haematoxylin and eosin or Masson’s trichrome according to standard protocols.

### 4.6. Immunofluorescence (IF)

The cryopreserved sections were blocked with 5% BSA/5% donkey serum (Sigma-Aldrich, St. Louis, MO, USA) and incubated overnight with primary antibodies: rat anti-mouse vimentin (1:300, Biolegend, San Diego, CA, USA) and rabbit anti-mouse SMA (1:400, Cell signalling). The slides were washed and incubated with Alexa Fluor-conjugated rat or rabbit secondary antibodies (1:200, BD Biosciences, Columbus, OH, USA). The whole graft area in each section was imaged on a Nikon Ti-E microscope. Staining was quantitated for integrated density using FIJI version 1.54f software (NIH). Thresholds were adjusted to minimise background using negative control staining.

### 4.7. Statistical Analysis

The data collected from the mice were grouped based on treatment option or days from the treatment, as specified in each figure. One-way or two-way ANOVA was performed, depending on the number of variables. An unpaired *t*-test was employed when comparing only two groups with significance * = *p* ≤ 0.05, ** = *p* ≤ 0.01, *** = *p* ≤ 0.001, **** = *p* ≤ 0.0001. Data were presented as mean +/− SEM (standard error of the mean).

## Figures and Tables

**Figure 1 ijms-24-16277-f001:**
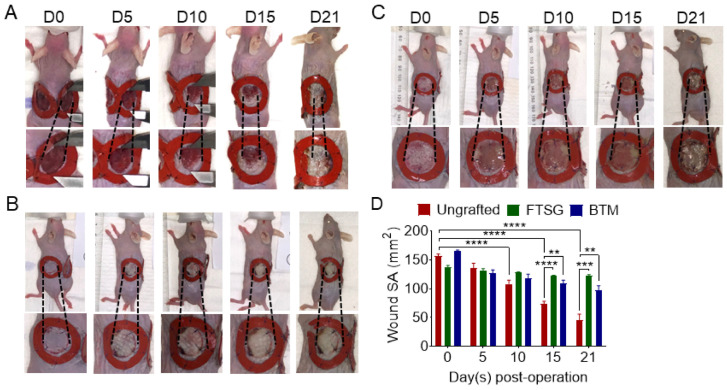
**Wound epithelisation/contraction.** Representative images of (**A**) ungrafted full-thickness wounds and wounds grafted with (**B**) FTSG or (**C**) BTM on day of surgery (D0), day 5 (D5), day 10 (D10), day 15 (D15), and day 21 (D21). (**D**) The wound surface area (SA) was measured at D0, D5, D10, D15, and D21 using ImageJ version 1.54f software. Data analysed using two-way ANOVA. Values represent mean +/− SEM in each group (*n* = 4 mice per group, ns = not significant, ** = *p* ≤ 0.01, *** = *p* ≤ 0.001, **** = *p* ≤ 0.0001).

**Figure 2 ijms-24-16277-f002:**
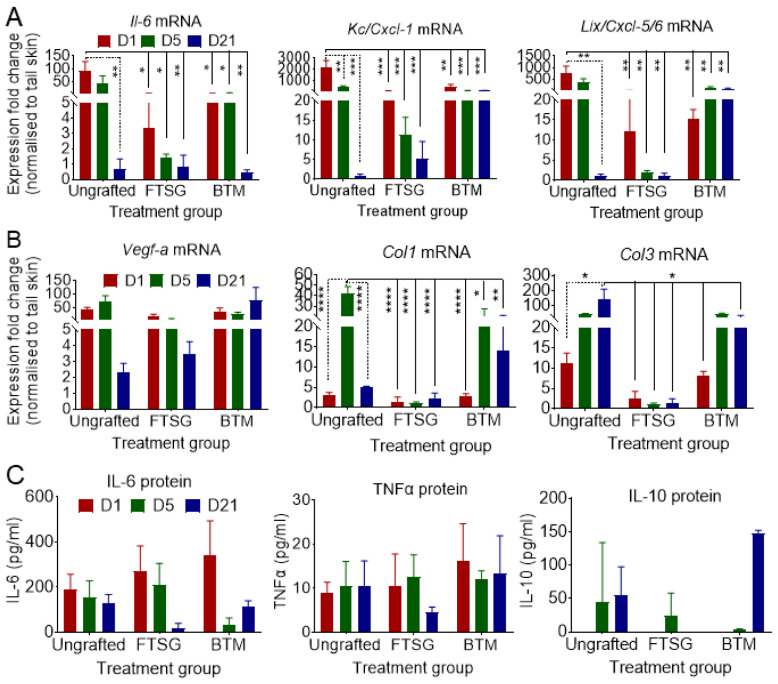
**Effect of skin grafting on inflammation and proliferation markers.** Graft tissue was harvested from ungrafted full-thickness wounds or wounds grafted with FTSG and BTM. Dermal tissue was separated at days 1, 5, and 21 post-operation. (**A**) inflammatory and (**B**) proliferative cytokine/growth factors were measured using RT-qPCR. Data were analysed as expression fold changes of the targets against the average of the C_t_ values of three housekeeping genes (*Polr2a*, *Eef1a1*, *Tuba1a*), and normalised against the C_t_ values of the same genes of the “tail” skin collected on the same day. (**C**) Protein isolated from ungrafted and FTSG/BTM-grafted wounds were analysed for IL-6, TNF-α, and IL-10 secretion using ELISA. Statistical analysis was performed using two-way ANOVA. Values represent mean +/− SEM in each group (*n* = 3 mice per group, ns = not significant, * = *p* ≤ 0.05, ** = *p* ≤ 0.01, *** = *p* ≤ 0.001, **** = *p* ≤ 0.0001).

**Figure 3 ijms-24-16277-f003:**
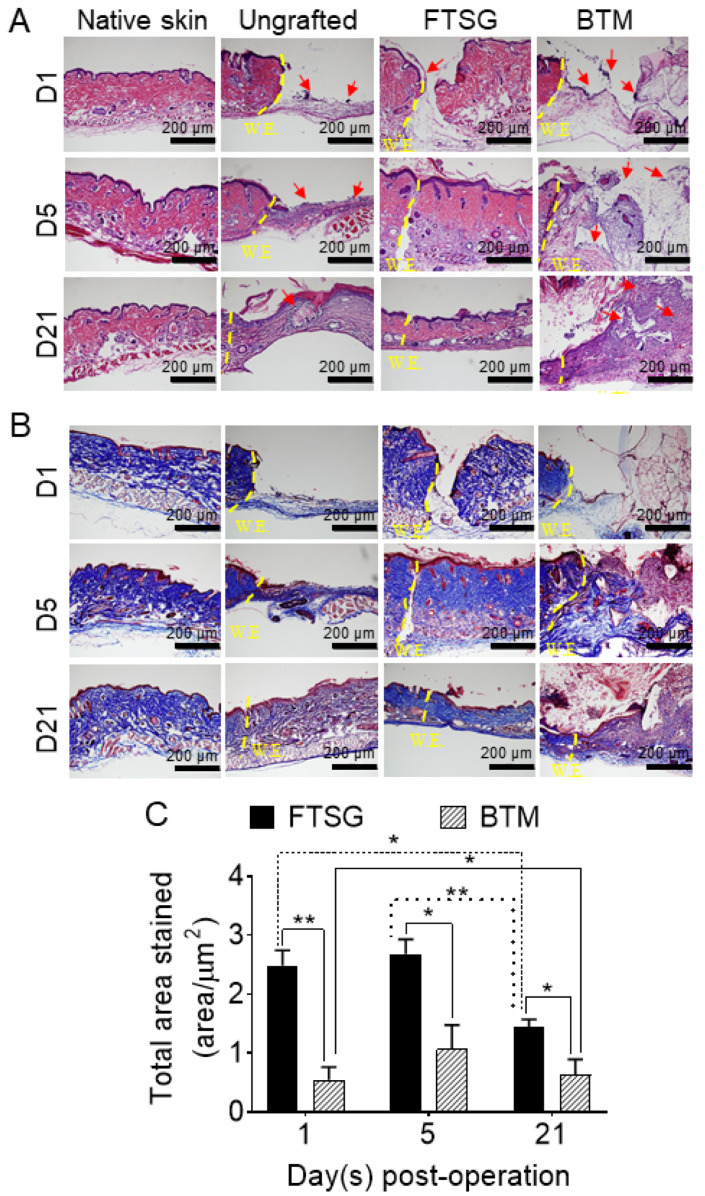
**Histological analysis of grafts.** Representative images of (**A**) haematoxylin and eosin and (**B**) Masson’s trichrome staining of the native skin prior to wounding, full-thickness ungrafted wounds, or wounds grafted with FTSG and BTM. Red arrows on haematoxylin and eosin images indicate the abundance of purple stained cells likely to be immune cells in BTM grafts. (**C**) Quantification of the total collagen area (area per µm^2^) that is stained blue with Masson’s trichrome at day 1, 5, and 21 post-grafting using ImageJ version 1.54f software. Statistical analysis was performed using one-way ANOVA. Values represent mean values +/− SEM in each group (*n* = 4–5 mice per group, * = *p* ≤ 0.05, ** = *p* ≤ 0.01).

**Figure 4 ijms-24-16277-f004:**
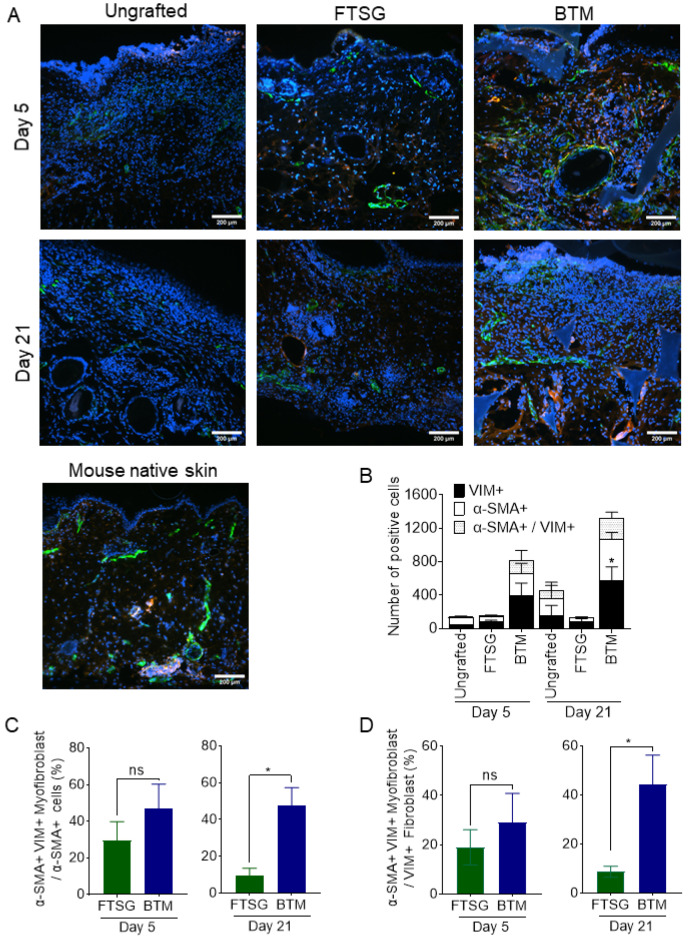
**Abundance of myofibroblasts in grafts.** Native skin prior to wounding, ungrafted full-thickness wounds, and wounds grafted with FTSG and BTM were co-stained for alpha-smooth muscle actin (α-SMA) in green and vimentin (VIM, a fibroblast marker) in red. Representative images of (**A**) α-SMA and VIM co-staining on day 5 and day 21 of FTSG and BTM grafts compared with native skin. Total number of cells expressing (**B**) VIM, α-SMA, and both (**C**) α-SMA^+^/VIM^+^ myofibroblast compared with the percentage of α-SMA^+^ cells and (**D**) α-SMA^+^/VIM^+^ myofibroblast compared with the percentage of VIM^+^ fibroblast were measured using NIS-elements version 5.21.00 software and normalised against no primary antibody negative control staining for each section. Data analysed using one-way ANOVA in (**B**) or an unpaired *t*-test in (**C**,**D**). Values represent mean +/− SEM in each group (*n* = 4 mice per group, ns = not significant, * = *p* ≤ 0.05).

**Figure 5 ijms-24-16277-f005:**
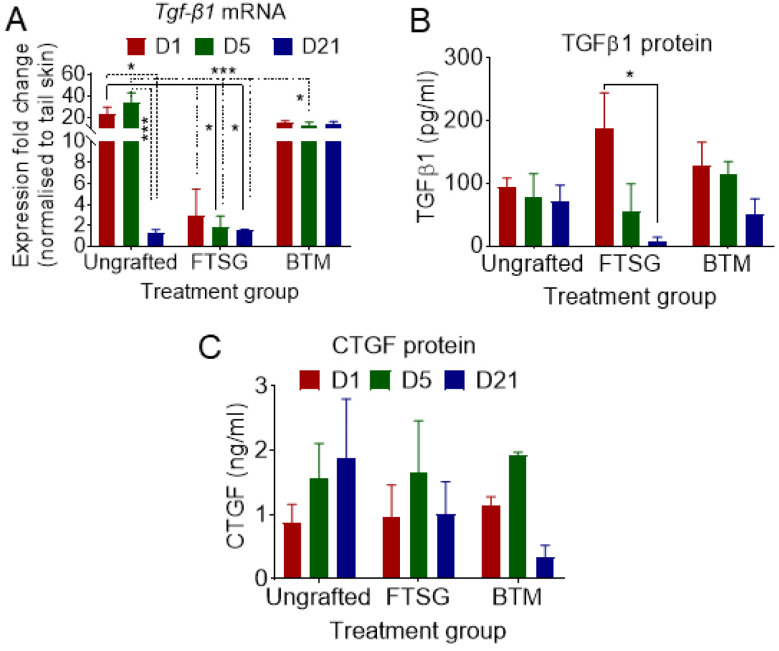
**Myofibroblast differentiation in BTM grafts is likely to be mediated at least partly by transforming growth factor beta 1 (TGF-β1).** (**A**) TGF-β1 mRNA was measured by RT-qPCR from the tissue of ungrafted full-thickness wounds or wounds grafted with FTSG and BTM on day 1, day 5, and day 21. Data were analysed as expression fold changes of the targets against the average of the C_t_ values of three housekeeping genes (*Polr2a*, *Eef1a1*, *Tuba1a*), and normalised against the C_t_ values of the same genes of the “tail” skin. Proteins isolated from ungrafted and FTSG/BTM-grafted wounds were analysed for (**B**) TGF-β1 and its downstream mediator (**C**) connective tissue growth factor (CTGF) secretion using ELISA. Data analysed using two-way ANOVA. Values represent mean +/− SEM in each group (*n* = 3–4 mice per group, * = *p* ≤ 0.05, *** = *p* ≤ 0.001).

**Table 1 ijms-24-16277-t001:** List of primers used in the study.

Gene of Interest	Reference	Forward Primer	Reverse Primer
*Il-1* *β*	[38]	5′-TTGACGGACCCCAAAAGATGAAG-3′	5′-TTCTCCACAGCCACAATGAG-3′
*Tnf-* *α*	[39]	5′-AGCCCACGTCGTAGCAAACCACCAA-3′	5′-ACACCCATTCCCTTCACAGAGCAAT-3′
*Il-6*	[40]	5′-ATGAAGTTCCTCTCTGCAAGAGACT-3′	5′-CACTAGGTTTGCCGAGTAGACTC-3′
*Tgf-* *β1*	[39]	5′-GCTAATGGTGGACCGCAACAACG-3′	5′-CTTGCTGTACTGTGTGTCCAGGC-3′
*Pdgf-bb*	In-house	5′-CGGAGTCGAGTTGGAAAGCTC-3′	5′-AATAACCCTGCCCACACTCT-3′
*Vegf-a*	[41]	5′-GCAGGCTGCTGTAACGATGAAG-3′	5′-GCTTTGGTGAGGTTTGATCCG-3′
*Il-10*	[42]	5′-ATTTGAATTCCCTGGGTGAGAAG-3′	5′-CACAGGGGAGAAATCGATGACA-3′
*Kc/Cxcl-1*	[43]	5′-ACTGCACCCAAACCGAAGTC-3′	5′-TGGGGACACCTTTTAGCATCTT-3′
*Lix/Cxcl-5/6*	In-house	5′-CTCGCCATTCATGCGGAT-3′	5′-AGCTTTCTTTTTGTCACTGCCC-3′
*Col3a1*	[44]	5′-TGGCACAGCAGTCCAACGTA-3′	5′-TGACATGGTTCTGGCTTCCA-3′
*Col1a1*	[45]	5‘-TCACCTACAGCACCCTTGTGG-3‘	5‘-CCCAAGTTCCGGTGTGACTC-3‘
*Eef1a1*	[46]	5′-GATGTTAGACGAGGCAATGTTG-3′	5′-CAATCCAGAACAGGAGCGTAG-3′
*Polr2a*	[47]	5′- CTCCTTTGAGGAAACGGTGGAT-3′	5′-GACTCCCTTCATCGGGTCACT-3′
*Tuba1a*	[48]	5′-CCTGGAACCCACGGTCATC-3′	5′-TGTAGTGGCCACGAGCATAGTTAT-3′

## Data Availability

Data are contained within the article or Appendix A.

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
