# Peer review of "Graft–Host Interaction and Its Effect on Wound Repair Using Mouse Models"

_ijms, 2023, doi:10.3390/ijms242216277_

Round 1

Reviewer 1 Report

Comments and Suggestions for Authors

The authors present an interesting comparison between the use of ungrafted, FTSG, and BTM to evaluate wound repair in a skin mouse model. Although the work is well-documented with key in vivo/in vitro assays, I have some doubts that I would like to have some clarification:

-       The first sentence of the abstract is confusing. Please improve it.

-       I recommend improving all the abstract. It is confusing and not clear. 

-       Why do you finalize the abstract with an aim/goal? This sentence shouldn’t be in the final part of the abstract but in the beginning of the abstract.  

-       There are acronymous without the meaning, namely in the introduction section.

-       Despite the detection and quantification of important anti-inflammatory and growth factors, did you detect the presence and amount of cells, namely macrophages and endothelial cells using specific biomarkers for these cells along the wound healing process?

Comments on the Quality of English Language

Minor editing of English language required.

Author Response

Please se the file attached.

Reviewer 2 Report

Comments and Suggestions for Authors

The article deals with repair in the skin after transplantation of autologous skin or a synthetic matrix. The study is exciting and offers conclusive insights. Unfortunately, the presentation of the data needs improvement. The results section (and probably the figures) needs to be restructured. There is constant jumping back and forth between figures in the text. This makes it difficult to follow the authors. Autologous skin is difficult to obtain, which is why synthetic matrixes are used. The authors must refer to this fact in their conclusion on possible meanings of their work. Furthermore information regarding statistics is completely missing in material and method section. In figure legends the authors state, that they use T-Tests which is inappropriate as more then two groups were compared

-        Abstract appears too long

-        It should be Eef1a1 instead of EF1A1

-        Murine genes should be written in lower case and italic. Protein should be capitalized and not italicized.

-        Figure legend of Fig.1 does not match the figure! C and D are missing

-        Why SEM instead of SD?

-        Why 4-5 sometimes ber also 3-7 mice per group? Please comment on that. How many mice were included in the study. This is not mentioned at all.

-        Single point representation should be performed in the graphs

-        Description of the figures is totally mixed up. Must be edited and rearranged, otherwise it is difficult to follow.

-        Align Fig. 4B with the others. Label scale bar in Fig. 4

-        Unpaired T-test not acceptable as a statistical test because more than two groups are being compared.

-        Only include p-values in the legends that actually occur.

-        "As previously described with some modification" must be specified. In general, methods must be presented in such a way that they become reproducible.

-        Specify dilution for ELISAs.

-        RNA was isolated from left, right and uninjured skin. Why is only one shown. Wouldn't it have made sense to use one wound for RNA and the other for histology to save animals.

-        How was the qPCR evaluated. Three reference genes were used. Of these, was the geomean used. In the given method only one control gene was mentioned.Please specify here.

-        Information about the primers should be added. How were they designed? Were they exon-intron spanning. Please specify the primers in a table.

Author Response

Please see file attached

Round 2

Reviewer 2 Report

Comments and Suggestions for Authors

The authors have implemented my points of criticism satisfactorily.